# Unified Analysis of Viscoelasticity and Viscoplasticity Using the Onsager Variational Principle

**DOI:** 10.3390/e27010055

**Published:** 2025-01-10

**Authors:** Kwang Soo Cho

**Affiliations:** Department of Polymer Science and Engineering, Kyungpook National University, Daegu 41566, Republic of Korea; polphy@knu.ac.kr

**Keywords:** Onsager variational principle, viscoelasticity, viscoplasticity, constitutive equation, internal variable

## Abstract

This study is the application of the Onsager variational principle to viscoelasticity and viscoplasticity with the minimization of the assumptions which are popularly used in conventional approaches. The conventional approaches assume Kröner–Lee decomposition, incompressible plastic deformation, flowing rule, stress equation and so on. These assumptions have been accumulated by many researchers for a long time and have shown many successful cases. The large number of successful assumptions leads to the conjecture that the mechanics can be described with a smaller number of assumptions. This paper shows that this conjecture is correct by using the Onsager variational principle.

## 1. Introduction

The electrode of the secondary battery is manufactured by coating slurry, which is a mixture of a large amount of active particle, a small amount of polymer adhesive and solvent fluid. Therefore, interest in the rheological properties of this slurry is increasing. In particular, the constitutive equation of the slurry is required for optimization of the coating process as well as rheological characterization [1].

Someone may consider the slurry as a viscoelastic fluid and others may consider it as a viscoplastic material or yield stress fluid [1]. Since metal under high stress level could be seen as a fluid because stress higher than yield stress give rise to the flow of metal. Therefore, viscoplasticity popular in metal plasticity would be applicable to the slurries of Li-ion battery. The polymer binder in the slurry may play a role in making the slurry behave like viscoelasticity, but its content is very low. A large number of active particles will maintain a certain cluster structure when the deformation is small, but when a certain level of stress is reached, the cluster structure will be destroyed and a large flow will be shown. The interaction between the active particles is expected to play a role in regenerating the cluster structure when the flow stops or weakens.

So how can we mathematically formulate such a complex mechanism for deformation? Such mathematical formalisms, called constitutive equations, can be said to be the result of accumulated experience and rationalization over many generations. The development method of the constitutive equation seems to have evolved along different paths depending on the type of material. In this paper, we focus on viscoelasticity and plasticity, which are highly related to slurry of Li-ion batteries. This is because the development method of these two classes of constitutive equation can include phenomena such as thixotropy and yield behavior [2,3,4], and these constitutive equations can be integrated into the thermodynamic theory of Cho and Lee [5].

The constitutive equations of linear elastic solids and linear viscous fluids can be said to be an intuitive formulation of the experimental results of the mechanical behavior of materials. However, it is hard to develop constitutive equations of viscoelasticity or viscoplasticity by use of such intuitive approach.

If we focus on the theory of finite deformation to deal with the general constitutive equation theory, the Kröner–Lee decomposition [6,7] can be said to be a monumental assumption in developing the constitutive equations of both viscoelasticity and viscoplasticity. In order for the Kröner–Lee decomposition to be used in the constitutive equation, a physical meaning must be given to the mathematically defined plastic velocity gradient. That is, constraints such as what the intermediate configuration should be and that the plastic velocity gradient should be a traceless symmetric tensor must be derived based on experimental results and imagination [8,9,10]. Then we arrive at the concept of the plastic deformation rate tensor $D_{p}$, and it is necessary to clarify how $D_{p}$ is related to a stress measure or to a function of elastic strain. The formulation of $D_{p}$ is called flow rule in plasticity theory and is deeply related with yield criterion. However, in the case of viscoelasticity, it is difficult to define a clear yield surface through experimental results, so $D_{p}$ must be formulated in a different way, application of viscosity model [4].

Most constitutive theories using thermodynamics apply the second law of thermodynamics to the formulation of $D_{p}$ for both viscoelasticity and viscoplasticity [8,11]. However, these theories cannot determine thermodynamically the stress equation which is the relation between stress and elastic strain. Additional assumption is demanded for the stress equation. An ad hoc assumption is used in the Leonov model of viscoelasticity [8]. It is remarkable that although not a thermodynamic method, the principle of virtual power was developed by Gurtin and co-workers [11,12,13,14] to derive the stress equation. However, this approach also demands several subsidiary assumptions.

In plasticity, hardening phenomena and Bauschinger effect are included in constitutive modeling without thermodynamics. These mechanical behaviors can be described by introduction of back stress and its evolution equation which was firstly modeled intuitively by Armstrong and Frederick [15].

Previous theories on the constitutive equation of viscoelasticity or plasticity consist of three parts: the relation between stress and elastic strain, the evolution equation of elastic strain and the flow rule. Construction of these three equations are based on a number of assumptions: principle of frame indifference, Kröner–Lee decomposition, incompressibility of intermediate configuration, hypothesis of co-directionality, non-negativity of entropy production rate and so on. Although the incompressibility of plastic deformation would be an example of experimentally confirmed hypothesis, the experimental evidence is confined to metallic material. To author’s knowledge, it would be hard to confirm the existence of the intermediate configuration in deforming polymeric material. At any way, it is true that most constitutive models based on such many assumptions works well in their applications. From the perspective that even complex natural phenomena are governed by a single law, if so many assumptions are valid, it would be possible to derive them from a smaller number of assumptions.

Could not these assumptions be replaced by a single thermodynamic theory? The purpose of this paper is to replace many of the seemingly thermodynamically independent assumptions and modeling used in the constitutive equation theory of viscoelasticity and viscoplasticity by the Onsager variational principle (OVP) alone. Thermodynamics of internal variable will be combined with OVP. Since Onsager’s paper [16] shows that it is difficult to apply OVP to viscoelasticity or viscoplasticity, there have been efforts to clarify whether the Onsager variational principle can be applied to such complex thermodynamic systems [5,17]. This paper will use the thermodynamics of Cho and Lee [5], but we will not deal with their detailed methodology here. We will simply use the consequences under the assumption that OVP is valid because Cho and Lee [5] showed the validity of OVP for the irreversible thermodynamics of internal variable.

## 2. Onsager Variational Principle

The Onsager variational principle is just the following variational equation:(1)$$\delta \int_{ℬ}\left(\psi -\rho \frac{ds}{dt}\right)dV=0,$$
where ℬ is the region where the material occupies in real space, ψ is the dissipation function, ρ is the mass density, *s* is the specific entropy (entropy per unit mass) of the system, d/dt is the material time derivative and d*V* is the volume element. Since the variational operator δ can be interpreted the functional differentiation of the integral, it is necessary to specify what the differentiation is with respect to and what the dissipation function is a function of. It is necessary to specify the general form of the evolution equations of the state variables in order to differentiate the integral.

There are various versions of OVP: use of Gibbs free energy [18]; use of Helmholtz free energy for isothermal process [19,20,21]. We prefer the OVP developed by Cho and Lee [5] which is not based on the linear relations between generalized fluxes and thermodynamic forces. The OVP of Cho and Lee [5] is briefly reviewed in Appendix A.

The specific entropy is assumed to depend on state variables such as the specific internal energy ε, the left Cauchy-Green strain $B\equiv F\cdot F^{T}$ and internal variables. Note that **F** is the deformation gradient. Internal variables could be a scalar, a vector or a tensor depending on the nature of the system under interest. We call ε and **B** classical state variables because they are the state variables of classical irreversible thermodynamics. We denote the arguments of the specific entropy by(2)$$S_{s}=\left(\varepsilon ,\;B,\;a,\;a,\;A\right),$$
where *a* is a scalar internal variable, **a** is a vector internal variable and **A** is a tensor internal variable. We call $S_{s}$ thermodynamic space for entropy. Depending on the nature of the system, one may choose the following thermodynamic space:(3)$$S_{s}=\left(\varepsilon ,\;B,\;A_{1},\;A_{2}\right),$$
where $A_{1}$ and $A_{2}$ are tensor internal variables.

Since we assume that internal variables represent the microscopic or mesoscopic structure of materials, it is natural that internal variables have some geometric features such that(4a)$$a=F\cdot \tilde{a},$$(4b)$$a=F^{-T}\cdot \tilde{a},$$(4c)$$A=F\cdot \tilde{A}\cdot F^{T},$$(4d)$$A=F^{-T}\cdot \tilde{A}\cdot F^{-1},$$
where **a** and **A** are internal variable at the current configuration and $\tilde{a}$ and $\tilde{A}$ are those at the reference configuration. Equation (4) is based on the assumption that the material deforms without any dissipative process. Note that a vector internal variable obeys Equation (4a) if it represents a microscopic structure like a fiber and it obeys Equation (4b) if it represents a microstructure like a surface. Equations (4c) and (4d) are tensor version of Equations (4a) and (4b). Scalar internal variables are considered as the magnitude of a vector internal variables or the invariants of a tensor internal variables. With the help of Equation (4), the material time derivative of internal variables without any dissipative process is given by(5)$$\begin{array}{l} {\left(\frac{da}{dt}\right)}_{\mathrm{kinematic}}=L\cdot a,\;{\left(\frac{da}{dt}\right)}_{\mathrm{kinematic}}=-L^{T}\cdot a,\\ {\left(\frac{dA}{dt}\right)}_{\mathrm{kinematic}}=L\cdot A+A\cdot L^{T},\;{\left(\frac{dA}{dt}\right)}_{\mathrm{kinematic}}=-L^{T}\cdot A-A\cdot L, \end{array}$$
where $L={\left(\nabla u\right)}^{T}$ is the velocity gradient. Here, we used dF/dt=L⋅F.

Since Equation (5) corresponds to the ideal case, the realistic evolution equation of internal variables should have the following form:(6)$$\begin{array}{l} \frac{da}{dt}={\left(\frac{da}{dt}\right)}_{\mathrm{kinematic}}+r-\mathrm{div}J,\\ \frac{dA}{dt}={\left(\frac{dA}{dt}\right)}_{\mathrm{kinematic}}+R-\mathrm{div}J, \end{array}$$
where **r** and **R** represent the effect of the dissipation process (or relaxation process) that occurs inside of material particle and divJ and divJ represent the dissipation process related with the interaction of a material particle with surrounding ones. It is obvious that **r** is a vector field, **R** is a second-order tensor field, **J** is a second-order tensor field and J is a third-order tensor field.

The evolution equations of classical state variables are given by conservation law and kinematic analysis as follows [22]:(7)$$\rho \frac{d\varepsilon }{dt}=-\mathrm{div}q+T:L,\;\frac{dB}{dt}=L\cdot B+B\cdot L^{T},$$
where **q** is heat flux and **T** is the Cauchy stress.

It is obvious that the material time derivatives of state variables should be zero at equilibrium. This equilibrium condition can be simply expressed by(8)F≡(q, L J, J, r, R)=(0, O, O, O, 0, O).
Therefore, the components of 𝔉 can be considered as the indicator of nonequilibrium because if at least one component of 𝔉 is nonzero, the system is in nonequilibrium. Cho and Lee [5] supposed that the dissipation function ψ should be a non-negative function of 𝔉:(9)ψ(q, L J, J, r, R)≥0.
The equality holds whenever Equation (8) holds. Furthermore, without loss of generality, the dissipation function could be dependent on state variables: $\psi \left(F,\;S_{s}\right)$. It is noteworthy that nonequilibrium indicators should be related with state variables. The relations are constitutive equations.

With the help of Equations (6) and (7), we know that ρds/dt is a linear functional of F because(10)$$\rho \frac{ds}{dt}=\rho \frac{\partial s}{\partial \varepsilon }\frac{d\varepsilon }{dt}+\rho \frac{\partial s}{\partial B}:\frac{dB}{dt}+\rho \frac{\partial s}{\partial A}:\frac{dA}{dt},$$
where for simplicity, we considered the case of $S_{s}=\left(\varepsilon ,\;B,\;A\right)$. Then Equation (1) can be replaced by(11)$$\frac{\delta }{\delta F}\int_{ℬ}\left(\psi -\rho \frac{ds}{dt}\right)dV=0.$$
Assuming that the variations of each component of 𝔉 are independent of each other, Equation (11) provides the following fundamental equations.(12a)$$\theta \frac{\partial \psi }{\partial L}=T+2\rho \theta B\cdot \frac{\partial s}{\partial B}+2\rho \theta A\cdot \frac{\partial s}{\partial A},$$(12b)$$\frac{\partial \psi }{\partial q}=\nabla \frac{1}{\theta },$$(12c)$$\frac{\partial \psi }{\partial J}=\nabla \frac{\partial s}{\partial A},$$(12d)$$\frac{\partial \psi }{\partial R}=\frac{\partial s}{\partial A},$$
where θ is the temperature defined as(13)$$\frac{\partial s}{\partial \varepsilon }=\frac{1}{\theta }>0.$$
Equation (12) implies that if the two scalar functions *s* and ψ are known then everything about constitutive equations are determined.

Cho and Lee [5] interpreted the physical meaning of dissipation function as the measure of how far a nonequilibrium state is from equilibrium. It is noteworthy that the product of temperature and entropy production rate is the dissipation used by most plasticians [9]. Therefore, they assumed that the dissipation function is a non-negative function of nonequilibrium indicators 𝔉 and state variables $S_{s}$ which satisfies Equation (9). Combination of Equations (10) and (11) with the evolution equations of state variables allows the interpretation of OVP such that any thermodynamic process is governed by the minimization of ψ constraint to the evolution of state variables. Therefore, the partial derivatives of entropy are the Lagrange multipliers and Equation (11) is the expression of the Lagrange multiplier method. This approach is very similar to the theory of Liu [23] although his aim is to rigorously determine entropy flux.

If the specific free energy *f* is defined as f=ε−θs then we can replace $S_{s}=\left(\varepsilon ,\;B,\;A\right)$ by $S_{f}=\left(\theta ,\;B,\;A\right)$ because(14)$$\frac{df}{dt}=-sd\theta +\frac{\partial f}{\partial B}:\frac{dB}{dt}+\frac{\partial f}{\partial A}:\frac{dA}{dt}.$$
Since $s=\theta^{-1}\left(\varepsilon -f\right)$, we can rewrite the OVP, Equation (1) by(15)$$\begin{array}{ll} 0 & =\delta \int_{ℬ}\left(\psi -\frac{\rho }{\theta }\frac{d\varepsilon }{dt}+\frac{\rho }{\theta }\frac{df}{dt}-\rho \left(\varepsilon -f\right)\frac{d\theta^{-1}}{dt}\right)dV\\ & =\delta \int_{ℬ}\frac{1}{\theta }\left[\theta \psi -\rho \frac{d\varepsilon }{dt}+\rho {\left(\frac{df}{dt}\right)}_{\theta }\right]dV, \end{array}$$
where ${\left(df/dt\right)}_{\theta }$ is the material time derivative of *f* at constant temperature. It is used that ∂f/∂θ=−s. Since the variational operator δ is independent of state variables such as θ, the free-energy version of OVP is given by(16)$$\delta \int_{ℬ}\left[\bar{\psi }-\rho \frac{d\varepsilon }{dt}+\rho {\left(\frac{df}{dt}\right)}_{\theta }\right]dV=0,$$
where $\bar{\psi }=\theta \psi$. It is noteworthy that although Doi successfully applied OVP to various problems in soft matter physics [19,20], his OVP differs from the original OVP because he omitted ρdε/dt. Hence, his OVP cannot provide stress equation such as Equation (12a) and he introduced an ad hoc remedy [24]. Appendix B briefly introduces the difference between Doi’s and our OVP.

Furthermore, Equation (16) provides the following fundamental equation for constitutive equation:(17a)$$\theta \frac{\partial \psi }{\partial L}=T-2\rho B\cdot \frac{\partial f}{\partial B}-2\rho A\cdot \frac{\partial f}{\partial A},$$(17b)$$\frac{\partial \psi }{\partial q}=\nabla \frac{1}{\theta },$$(17c)$$\frac{\partial \psi }{\partial J}=-\nabla \frac{1}{\theta }\frac{\partial f}{\partial A},$$(17d)$$\frac{\partial \psi }{\partial R}=-\frac{1}{\theta }\frac{\partial f}{\partial A}.$$
It is noteworthy that Equation (17) can also be obtained from Equation (12) by the consequences of the Legendre transform such that $\partial s/\partial \chi =-\theta^{-1}\partial f/\partial \chi$, where χ is a state variable of $S_{s}$ except ε. Although the existence of J is important for computational stability [25], most previous constitutive models do not include J, we consider only the case without J. It is also noteworthy that Equation (17) holds for non-isothermal process, too.

Before going into detail about how to develop the viscoelastic or viscoplastic constitutive equations using Equations (12) or (17), it is necessary to discuss how state variable **B** is treated in fluids. According to kinematic analysis it is obvious that(18)$$\frac{\tilde{\rho }}{\rho }=\det F=\sqrt{\det B},$$
where $\tilde{\rho }$ is the mass density at the reference configuration. Since the specific volume is the reciprocal of the mass density, the following holds true:(19)$$v=\tilde{v}\sqrt{\det B},\;\tilde{v}\equiv \frac{1}{\tilde{\rho }}.$$
Therefore, if the specific free energy depends only on detB, we can replace **B** in the list of state variables by the specific volume *v*, and Equations (12a) and (17a) become:(20)$$T=-pI+2\rho A\cdot \frac{\partial f}{\partial A}+\theta \frac{\partial \psi }{\partial L},$$
where, **I** is the identity tensor and(21)$$-2\rho B\cdot \frac{\partial f}{\partial B}=pI,\;p=-\frac{\partial f}{\partial v}.$$

Incompressibility is a useful approximation of mechanical behavior. The OVP can include incompressibility by adding $\hat{p}\mathrm{tr}L$ to the integrand of Equation (1) or Equation (16):(22)$$\delta \int_{ℬ}\left[\bar{\psi }-\rho \frac{d\varepsilon }{dt}+\rho {\left(\frac{df}{dt}\right)}_{\theta }+\hat{p}\mathrm{tr}L\right]dV=0.$$
As for incompressible fluid, it can be assumed that f=f(θ, A). Then Equation (22) results in(23)$$T=-\hat{p}I+2\rho A\cdot \frac{\partial f}{\partial A}+\theta \frac{\partial \psi }{\partial L}.$$
Of course, $\hat{p}$ is the hydrostatic pressure which can be determined by the boundary condition of the thermodynamic system under interest.

It is a reasonable assumption that the dissipation function ψ is an isotropic scalar function. Then it is obvious that ψ is dependent on $D=\frac{1}{2}\left(L+L^{T}\right)$ rather than **L** and that we can write(24)$$\frac{\partial \psi }{\partial L}=\frac{\partial \psi }{\partial D}.$$
Equation (24) guarantees the symmetry of stress under the assumption that the tensor internal variable **A** is a symmetric tensor. We can interpret θ∂ψ/∂L as a viscous stress which may be negligible for most polymeric melts.

## 3. Replacement of Multiplicative Decomposition

Leonov [26] showed that a number of constitutive equations of viscoelastic fluid consist of two equations: the evolution equation of conformation tensor, here tensor internal variable and the stress equation:(25a)$$\frac{dA}{dt}=L\cdot A+A\cdot L^{T}+R,$$(25b)T=−pI+S,
where **R** and **S** are tensor-valued functions of **A**. It is basically assumed that **A** is a positive definite and symmetric tensor. The symmetry of **A** immediately implies that **R** is also a symmetric tensor. Since in the limit of R→O the evolution equation of **A** is identical to that of $B=F\cdot F^{T}$ which is a symmetric and positive definite tensor, it is a reasonable assumption that **A** is a symmetric and positive definite. Note that Equation (6b) becomes the first equation of Equation (25) if J=O.

The Phan-Thien and Tanner model (PTT model) [27,28] and the Giesekus model [29] are based on simple molecular picture that gives Equation (25a) and specifies the function structure of **R**. On the other hand, the Leonov model uses the Kröner–Lee decomposition to obtain Equation (25a). However, the Kröner–Lee decomposition cannot specify the functional structure of **R**. Leonov applied irreversible thermodynamics to the determination of **R**. Leonov determined the functional form of **R** under the condition that **R** only requires that the entropy production rate not be negative. This condition agrees with Equation (17d).

### 3.1. Approach Based on Kröner–Lee Decomposition

Since one of the purposes of this paper is to replace the Kröner–Lee decomposition by thermodynamic principle, it is necessary to review briefly the Kröner–Lee decomposition. The Kröner–Lee decomposition considers tensor internal variable as elastic strain:(26)$$A=F_{e}\cdot F_{e}^{T},$$
where $F_{e}$ is the elastic part of deformation gradient. The deformation gradient **F** is decomposed multiplicatively:(27)$$F=F_{e}\cdot F_{p},$$
where $F_{p}$ is the plastic part of deformation gradient. Since the velocity gradient is given by $L=dF/dt\cdot F^{-1}$, one may define elastic velocity gradient as(28)$$L_{e}=\frac{dF_{e}}{dt}\cdot F_{e}^{-1}.$$
Then velocity gradient can be expressed in terms of $F_{e}$ and $F_{p}$ as follows(29a)$$L=L_{e}+L_{p},$$(29b)$$L_{p}\equiv F_{e}\cdot {\hat{L}}_{p}\cdot F_{e}^{-1},$$(29c)$${\hat{L}}_{p}=\frac{dF_{p}}{dt}\cdot F_{p}^{-1}.$$
According to Gurtin, Fried and Anand [9], **L**, $L_{e}$ and $L_{p}$ are spatial tensors while ${\hat{L}}_{p}$ is a structural tensor. Spatial tensor is a linear transform from spatial vector to spatial vector and structural tensor is a linear transform from structural vector to structural vector. Spatial vector is a vector defined on the current configuration and structural vector is a vector defined on the intermediate configuration. Similarly, one may define material vector and material tensor as those on the reference configuration.

Since $L_{e}=L-{\hat{L}}_{p}$, the evolution equation of **A** is given by(30)$$\frac{dA}{dt}=L_{e}\cdot A+A\cdot L_{e}^{T}=L\cdot A+A\cdot L^{T}-L_{p}\cdot A-A\cdot L_{p}^{T}.$$
Comparison of Equation (30) with Equation (25) gives(31)$$R=-L_{p}\cdot A-A\cdot L_{p}^{T}.$$
With the help of Equation (29b), we can rewrite Equation (31) as follows(32)$$R=-2F_{e}\cdot {\hat{D}}_{p}\cdot F_{e}^{T},\;{\hat{D}}_{p}=\frac{1}{2}\left({\hat{L}}_{p}+{\hat{L}}_{p}^{T}\right),\;{\hat{L}}_{p}\equiv \frac{dF_{p}}{dt}\cdot F_{p}^{-1}.$$
Similarly, one may prefer spatial tensor $D_{p}=\frac{1}{2}\left(L_{p}+L_{p}^{T}\right)$ to structural tensor ${\hat{D}}_{p}$.

Leonov [8] showed that if material is incompressible then two among the followings are independent:(33)$$\mathrm{tr}L=0,\;\mathrm{tr}L_{p}=0,\;\det A=1.$$
The condition $\mathrm{tr}L_{p}=0$ is equivalent to $\mathrm{tr}{\hat{L}}_{p}=0$ and implies that the intermediate configuration has the same mass density of the reference configuration. In metal plasticity, it is assumed that plastic deformation does not generate any volume change [9]. For the uniqueness of the Kröner–Lee decomposition, it is assumed that any rotation is focused on $F_{e}$ [9,14]. Then both $L_{p}$ and ${\hat{L}}_{p}$ should be symmetric. Leonov also accepted the symmetry of $L_{p}$ and derived(34)$$R=-2A\cdot D_{p},\;D_{p}=\frac{1}{2}\left(L_{p}+L_{p}^{T}\right),$$
Note that thanks to the thermodynamics, Leonov revealed that $D_{p}$ is coaxial with **A** and is proportional to the deviatoric part of **S**:(35)$$D_{p}=\frac{1}{\eta_{p}}S^{\prime },\;S^{\prime }=S-\frac{\mathrm{tr}S}{3}I,$$
where the viscosity $\eta_{p}$ is a scalar function of state variables and $D=\frac{1}{2}\left(L+L^{T}\right)$. The argument of the viscosity depends on the material class such as viscoelasticity and viscoplasticity.

Because of Equation (17d), it is obvious that **R** is a tensor-valued function of **A**. Since **A** is a symmetric and positive definite tensor, it must be invertible. Therefore, if the thermodynamic principle determines **R** then we can define plastic deformation rate by $D_{p}=-\frac{1}{2}A^{-1}\cdot R$. This thermodynamic definition of plastic deformation rate implies that $D_{p}$ is symmetric because both $A^{-1}$ and **R** are symmetric and **A** and **R** commute: A⋅R=R⋅A. Theories based on the Kröner–Lee decomposition require the assumption that $L_{p}$ must be symmetric, and the basis for this must be found in the properties of the intermediate configuration (such as isotropy). However, the thermodynamic definition of $D_{p}$ can omit such a cumbersome reasoning process. This thermodynamic definition can be said to be more realistic, especially when dealing with materials such as polymer materials, where it is difficult to experimentally identify the intermediate configuration.

Since plasticians prefer ${\hat{D}}_{p}$ to $D_{p}$, their flowing rule is to relate ${\hat{D}}_{p}$ by a stress measure which is a structural tensor. The new stress measure is called the Mandel stress ${\hat{M}}_{e}$ which is related with the Cauchy stress which is a spatial tensor [9]. A push forward operation for the flowing rule ${\hat{D}}_{p}={\hat{M}}_{e}/{\hat{\eta }}_{p}$ results in Equation (35).

### 3.2. Non-Thermodynamic Approach Without Kröner–Lee Decomposition

The Kröner–Lee decomposition is not the only way to construct the evolution equation of **A**. Examples are the Phan-Thien and Tanner model (PTT model) [27,28] and the Giesekus model [29]:(36)$$\lambda R=\{\begin{array}{ll} -e^{\beta \left(\mathrm{tr}A-3\right)}\left(A-I\right) & \mathrm{PTT}\\ -\beta A^{2}-\left(1-2\beta \right)A-\left(\beta -1\right)I & \mathrm{Giesekus} \end{array},$$
where β is the nonlinear material constant and λ is the relaxation time of linear viscoelastic regime. The derivation of **R** of the PTT model is based on the kinetics of temporary network and that of the Giesekus model is based on the anisotropic drag of polymer chain. These models assume simple picture of molecular structures of polymeric fluids. Although the approaches based on the Kröner–Lee decomposition demands the modeling of the function structure of **R**, an approach based on a simple molecular picture often provides Equation (25a). However, the stress equation, Equation (25b) is developed from another basis. It is because momentum balance equation is not derived from the kinetics of the simple picture of molecular structure [30,31]. It is the motivation of dissipative particle dynamics (DPD) that the original Fokker-Planck equation cannot preserve momentum balance equation.

Since **A** is assumed as a positive definite and symmetric tensor, Equation (36) allows the plastic deformation rate tensor $D_{p}$ as follows:(37)$$2\lambda D_{p}=\{\begin{array}{ll} e^{\beta \left(\mathrm{tr}A-3\right)}\left(I-A\right) & \mathrm{PTT}\\ \beta A+\left(1-2\beta \right)I+\left(\beta -1\right)A^{-1} & \mathrm{Giesekus} \end{array},$$
If Equation (17d) determines **R** as a tensor-valued function of **A** then we can determine the plastic deformation rate tensor $D_{p}$ without loss of generality. If we adopt the general form of **R** by Equation (34) then Equation (25a) will have a consistent form:(38)$$\frac{dA}{dt}=L\cdot A+A\cdot L^{T}-D_{p}\cdot A-A\cdot D_{p}.$$
Note that Equation (38) is based on that $D_{p}$ is coaxial: $D_{p}\cdot A=A\cdot D_{p}$, which will be given from the modeling of the dissipation function.

### 3.3. Thermodynamic Approach Without Kröner–Lee Decomposition

Since Equation (17b) is the constitutive equation of heat flux, we assume that the dissipation function has the following form:(39)$$\psi \left(q,\;L,\;R,\;S\right)=\psi_{q}\left(q,\;S\right)+\psi_{\mathrm{rlx}}\left(R,\;\theta ,\;A\right)\geq 0,$$
where S=(θ, A) represents the state variables which are the arguments of the specific free energy. In order to find **R** in terms of **A** and θ, we have to solve(40)$$\theta \frac{\partial \psi_{\mathrm{rlx}}}{\partial R}=-\rho \frac{\partial f}{\partial A}.$$
To make the mathematical problem easier, we assume that $\psi_{\mathrm{rlx}}$ is an isotropic quadratic function of **R**:(41)$$\psi_{\mathrm{rlx}}\left(Q\cdot R\cdot Q^{T},\;Q\cdot A\cdot Q^{T},\;\theta \right)=\psi_{\mathrm{rlx}}\left(R,\;A,\;\theta \right),$$
and(42)$$\psi_{\mathrm{rlx}}\left(R,\;A,\;\theta \right)=\frac{1}{2}R:M\left(A,\;\theta \right):R=\frac{1}{2}M_{ikpq}\left(A,\;\theta \right)R_{ik}R_{pq},$$
where $M=M_{ikpq}e_{i}⊗e_{k}⊗e_{p}⊗e_{q}$ is a positive definite tensor of fourth order. Here, **Q** is an arbitrary orthogonal tensor such that $Q\cdot Q^{T}=Q^{T}\cdot Q=I$ and detQ=1. It is not easy to construct 𝕄 in terms of **A** with making dissipation function satisfy Equation (41), although a trial for seeking the general form of M is found in Leonov [8]. Since temperature dependence of $\psi_{\mathrm{rlx}}$ is not important in construction of isotropic function, we omit θ in the expression of dissipation function for simplicity.

One of the simplest ways to make a non-negative quadratic function of **R** with $A^{m}$ would be(43)$$\phi_{m}\left(A,\;R\right)=a_{m}{\left(\mathrm{tr}A^{m}\cdot R\right)}^{2}+b_{m}\left(A^{m}\cdot R\right):\left(A^{m}\cdot R\right)\geq 0,$$
where $a_{m}>0$ and $b_{m}>0$ are scalar-valued function of the invariants of **A** and *m* is an integer. Note that for any second order tensor **Z**, it is obvious that ${\left(\mathrm{tr}Z\right)}^{2}\geq 0$ and Z:Z≥0. The two equalities hold simultaneously whenever Z=O. For any integer *m*, $A^{m}$ can be represented by a linear combination of **I**, **A** and $A^{2}$ thanks to the Cayley-Hamilton theorem such that(44)$$A^{3}-I_{A}A^{2}+II_{A}A-III_{A}I=O,$$
where $I_{A}$, $II_{A}$ and $III_{A}$ are the principal invariants of **A**:(45)$$I_{A}\equiv \mathrm{tr}A;\;II_{A}\equiv \frac{{\left(\mathrm{tr}A\right)}^{2}-\mathrm{tr}A^{2}}{2};\;III_{A}\equiv \det A.$$
Therefore, one may suggest the following dissipation function(46)$$\psi_{\mathrm{rlx}}\left(A,\;R\right)=\sum_{m=0}^{2}\phi_{m}\left(A,\;R\right).$$
Since the conformation tensor **A** is considered as a symmetric and positive definite tensor, it is invertible and one may prefer(47)$$\psi_{\mathrm{rlx}}\left(A,\;R\right)=\sum_{m=-1}^{1}\phi_{m}\left(A,\;R\right).$$

We suggest a simpler construction of dissipation function such as(48)$$\psi_{\mathrm{rlx}}\left(A,\;R\right)=\frac{\mu_{1}}{2}{\left(\mathrm{tr}G\cdot R\right)}^{2}+\frac{\mu_{2}}{2}\left(G\cdot R\right):\left(G\cdot R\right)\geq 0,$$
where $\mu_{1}$ and $\mu_{2}$ are positive functions of the invariants of **A** and **G** is a tensor-valued function of **A** such that(49)$$G=c_{0}I+c_{1}A+c_{2}A^{-1},$$
where $c_{0}$, $c_{1}$ and $c_{2}$ are functions of the invariants of **A** and θ.

Application of Equation (49) to Equation (40) gives(50)$$\mu_{1}\left(\mathrm{tr}G\cdot R\right)G+\mu_{2}G^{2}\cdot R=-\frac{\rho }{\theta }\frac{\partial f}{\partial A}.$$
We assume that **G** is invertible. Then Equation (50) can be rewritten by(51)$$\mu_{1}\left(\mathrm{tr}G\cdot R\right)I+\mu_{2}G\cdot R=-\frac{\rho }{\theta }G^{-1}\cdot \frac{\partial f}{\partial A}.$$
To solve Equation (50) for **R**, we take trace on both sides of Equation (51) and have(52)$$\mathrm{tr}G\cdot R=-\frac{\rho }{\theta \left(3\mu_{1}+\mu_{2}\right)}\mathrm{tr}G^{-1}\cdot \frac{\partial f}{\partial A}.$$
Substitution of Equation (52) to Equation (51) and rearrangement give(53)$$R=-\alpha G^{-1}\cdot \left[\hat{S}-\tilde{\zeta }\left(\mathrm{tr}\hat{S}\right)I\right],\;\hat{S}\equiv 2G^{-1}\cdot \rho \frac{\partial f}{\partial A},$$
where(54)$$\alpha =\frac{1}{2\theta \mu_{2}};\;\tilde{\zeta }=\frac{\mu_{1}}{3\mu_{1}+\mu_{2}}.$$
Note that $\hat{S}$ is not a structural tensor of Gurtin, Fried and Anand [9] because our approach does not have to rely on any theoretical tool based on the Kröner–Lee decomposition. The dimensionless parameter $\tilde{\zeta }$ must be positive and less than unity.

If $\mu_{1}\gg \mu_{2}$ then $\tilde{\zeta }\approx \frac{1}{3}$ and Equation (53) can be rewritten by(55)$$R=-\alpha G^{-1}\cdot \hat{S^{\prime }},\;\hat{S^{\prime }}\equiv \hat{S}-\frac{\mathrm{tr}\hat{S}}{3}I.$$
We know that $\hat{S^{\prime }}$ is the deviatoric tensor of $\hat{S}$. With the help of Equations (23) and the assumption of no viscous stress, the extra stress **S** is given by(56)$$S=2\rho A\cdot \frac{\partial f}{\partial A}.$$
Then we can express $\hat{S}$ in terms of extra stress **S**:(57)$$\hat{S}=G^{-1}\cdot A^{-1}\cdot S.$$
Substitution of Equation (57) to Equation (53) yields(58)$$R=-\alpha G^{-1}\cdot \left[G^{-1}\cdot A^{-1}\cdot S-\tilde{\zeta }\left(\mathrm{tr}G^{-1}\cdot A^{-1}\cdot S\right)I\right].$$

One of the simplest cases, one may think of $G=A^{-1}$. Then Equation (58) becomes(59)$$R=-\alpha A\cdot \left[S-\tilde{\zeta }\left(\mathrm{tr}S\right)I\right].$$
If $\tilde{\zeta }\approx \frac{1}{3}$ then Equation (59) is very similar to Equations (34) and (35) which are the relaxation term of the constitutive models based on the Kröner–Lee decomposition such as the Leonov model. The Leonov model derives the plastic deformation tensor is a deviatoric tensor and proportional to extra stress **S**, which agrees with Equation (39). Note that comparison of Equation (35) with Equation (59) yields(60)$$D_{p}=\frac{\alpha }{2}S^{\prime },\;\alpha =\frac{2}{\eta_{p}}.$$
Therefore, it can be said that we derived the Leonov model without the Kröner–Lee decomposition and a number of subsidiary assumptions.

Consider a model of viscoelastic fluid with the simplest free energy model such as(61)$$S=2\rho A\cdot \frac{\partial f}{\partial A}=G_{o}\left(A-I\right),$$
where $G_{o}>0$ is a material constant corresponding to modulus (relaxation intensity of linear viscoelasticity). Substitution of Equation (61) to Equation (58) yields(62)$$R=-\alpha G_{o}G^{-1}\cdot \left[G^{-1}\cdot \left(I-A^{-1}\right)-\tilde{\zeta }\left(\mathrm{tr}G^{-1}\cdot \left(I-A^{-1}\right)\right)I\right].$$
Under the assumption that $\mu_{1}=0$, Equation (62) is reduced to(63)$$R=-\alpha G_{o}G^{-2}\cdot \left(I-A^{-1}\right).$$
Since conformation tensor is assumed symmetric and positive definite, it is obvious that(64)$$A=a_{1}^{2}{\hat{a}}_{1}⊗{\hat{a}}_{1}+a_{2}^{2}{\hat{a}}_{2}⊗{\hat{a}}_{2}+a_{3}^{2}{\hat{a}}_{3}⊗{\hat{a}}_{3},$$
where $a_{k}^{2}>0$ are eigenvalues of **A** and ${\hat{a}}_{k}$ are normalized eigenvectors of **A**. Since ${\hat{a}}_{i}\cdot {\hat{a}}_{k}=\delta_{ik}$, it is straightforward that we can define(65)$$\begin{array}{l} A^{1/2}=a_{1}{\hat{a}}_{1}⊗{\hat{a}}_{1}+a_{2}{\hat{a}}_{2}⊗{\hat{a}}_{2}+a_{3}{\hat{a}}_{3}⊗{\hat{a}}_{3};\\ A^{-1/2}=\frac{1}{a_{1}}{\hat{a}}_{1}⊗{\hat{a}}_{1}+\frac{1}{a_{2}}{\hat{a}}_{2}⊗{\hat{a}}_{2}+\frac{1}{a_{3}}{\hat{a}}_{3}⊗{\hat{a}}_{3}. \end{array}$$
If we choose $G=A^{-1/2}$ then we have $G^{-1}=A^{1/2}$ and Equation (63) becomes(66)$$R=-\alpha G_{o}\left(A-I\right)=-\alpha S.$$
Note that α has the unit corresponding to the reciprocal of viscosity and $G_{o}$ corresponds to the relaxation intensity in the linear viscoelastic regime. Then, using the following definitions, this equation becomes the PTT model:(67)$$\alpha \left(A,\;\theta \right)=\frac{\exp\left[\beta \left(\mathrm{tr}A-3\right)\right]}{\lambda \left(\theta \right)G_{o}\left(\theta \right)}.$$
Of course, λ(θ) is the relaxation time of linear viscoelasticity.

Consider the case of $\tilde{\zeta }=0$ and Equation (61) again. Suppose that(68)$$G^{-2}=\frac{\beta A\cdot \left(A-I\right)+A}{\lambda \alpha G_{o}}.$$
Substitution of Equation (68) to Equation (63) gives(69)$$R=-\frac{1}{\lambda }\left[\beta A^{2}+\left(1-2\beta \right)A+\left(\beta -1\right)I\right].$$
This is the Giesekus model.

It is not essential to use the hypothetical decomposition of Kröner and Lee in the construction of viscoelastic constitutive equation because the evolution equation of conformation tensor based on the Kröner–Lee decomposition is one of possible formulations which can be obtained from the variational thermodynamics of Cho and Lee [5]. However, the idea of the Kröner–Lee decomposition is invaluable in description of kinematic hardening or thixotropy, which involve structural changes which are believed to be affected mainly by plastic deformation. Therefore, we will focus on only the formulation of **R** which agrees with the Kröner–Lee decomposition.

## 4. Need for New Internal Variable

The 1D analog of the constitutive modeling in Section 3 is a connection of elastic and inelastic elements in series as shown in Figure 1a. The inelastic element includes both viscous and plastic dashpots. Plastic dashpot does not generate any plastic deformation for unloading and not yielding. Furthermore, plastic dashpot is independent of how deformation is fast. The condition of rate-independence makes the fluidity $\alpha =1/\eta_{p}$ of plastic dashpot proportional to deformation rate **D**. Since this means that if D=O then $D_{p}=O$, no stress relaxation is observed at a relaxation test. That is, when an arbitrary load is applied and the deformation is fixed and the stress is measured, the stress remains the same as when the deformation was fixed.

On the other hand, the viscous dashpot can be linear or nonlinear, but is always independent of the yield or load conditions. This means that $D_{p}$ is independent of **D** and is a function of only stress such that $D_{p}=O$ whenever S=O. Note that S=O is the case that the conformation tensor has the value of equilibrium, A=I. Since stress is a function of conformation tensor **A** (or elastic strain), it can be said that $D_{p}$ is a function of **A**. Therefore, the model based on viscous dashpot shows stress relaxation and the fully relaxed stress is always zero tensor. Therefore, any 3D model analogous to the 1D model of Figure 1a with viscous dashpot cannot describe viscoelastic solid which has non-zero fully relaxed stress. To make a realistic model for viscoelastic solid, the 2-element model of Figure 1a is needed to be connected with an elastic element in parallel as shown in Figure 1b. If the dashpot of 3-element model I is viscous then the fully relaxed stress of the model is $T_{2}$ and the total stress **T** should be the sum of $T_{1}$ and $T_{2}$. The two stresses $T_{1}$ and $T_{2}$ are functions of internal variables $A_{1}$ and $A_{2}$, respectively, but the model does not imply that **T** is a function of $A_{1}$ and $A_{2}$.

In plasticity, the 3-element model II represented by Figure 1c can describe kinematic hardening if the dashpot is a plastic one. It is interesting that the internal variable $A_{b}$ does not affect the stress **T** in a direct manner but the evolution of $A_{b}$ determines the back stress $T_{b}$ and the difference between **T** and $T_{b}$ controls the plastic deformation rate $D_{p}$. It is noteworthy that Arruda and Boyce [32] applied the 3-element model II to amorphous polymer solid with the back stress $T_{b}$ which obeys a nonlinear rubber elastic model (Langevin spring) and the total stress **T** which obeys a linear elasticity.

As most plasticians do, Arruda and Boyce [32] defined the deformation of linear and nonlinear springs using the Kröner–Lee decomposition: $U_{e}=\sqrt{F_{e}^{T}\cdot F_{e}}$ and $V_{b}=\sqrt{F_{p}\cdot F_{p}^{T}}$, respectively. Note that $U_{e}$ and $V_{p}$ correspond to **A** and $A_{b}$ of Figure 1c, respectively. Their stress equation is assumed as(70)$$T=\frac{1}{J}G_{e}:\log U_{e},\;J=\det F_{e},$$
where $G_{e}$ is the modulus tensor of linear elasticity, which is a positive definite and symmetric tensor of fourth order. The plastic deformation rate $D_{p}$ of their dashpot is assumed to obey(71)$$D_{p}={\dot{\gamma }}_{p}N,\;N=\frac{1}{\tau }\Delta T,\;\tau =\sqrt{\frac{1}{2}\Delta T:\Delta T},$$
where(72)$$\Delta T=T-T_{b},\;T_{b}=\frac{1}{J}F_{e}\cdot {\hat{T}}_{b}\cdot F_{e}^{T},\;{\hat{T}}_{b}={\hat{T}}_{b}\left(V_{p}\right).$$
and ${\dot{\gamma }}_{p}$ is the plastic shear rate of the Argon model [33], which is a function of τ. Equation (71) is the mathematical representation of co-direction hypothesis that plastic deformation rate should have the same direction of the deriving stress. The model is the combination of commonly accepted principles (assumptions) in plastic mechanics and Argon’s molecular model for the yielding of polymer solids. Since the evolution of $U_{e}$ determines that of $V_{p}$ according to the Kroner-Lee decomposition, their model does not have to model the evolution equation of back stress. However, the most widely used theory dealing with kinematic hardening is the evolution equation of back stress proposed by Armstrong and Frederick [15]. Therefore, our thermodynamic approach needs the evolution equation of $A_{b}$ as well as that of **A**.

The lesson from Section 3 allows us to use the evolution equation of **A** such that(73)$$\frac{dA}{dt}=L\cdot A+A\cdot L^{T}-2A\cdot D_{p}.$$
Our thermodynamic theory determines $D_{p}$ as a function of **A** through free energy and existence of **L** in the right-hand side of Equation (73) makes the true stress contains the term 2ρA⋅∂f/∂A. We assume that the evolution of the internal variable $A_{b}$, which is related with back stress, obeys the following equation:(74)$$\frac{dA_{b}}{dt}=D_{p}\cdot A+A\cdot D_{p}-2A\cdot D_{b}.$$
We replace **A** and **L** in Equation (73) by $A_{b}$ and $D_{p}$, respectively, and introduce $D_{b}$ to represent the relaxation of the new internal variable. Since the evolution equation, Equation (74) does not contain L, the stress equation will not have a tensor-valued function of $A_{b}$. Since we call **A** elastic strain (or conformation tensor for polymeric materials), we can call $A_{b}$ back strain.

If we choose thermodynamic space as $S_{s}=\left(\varepsilon ,\;A,\;A_{b}\right)$, OVP with the evolution equations, Equations (73) and (74) provides(75)$$\begin{array}{l} \frac{\partial \psi }{\partial L}-\frac{1}{\theta }T-2\rho A\cdot \frac{\partial s}{\partial A}=O,\\ \frac{\partial \psi }{\partial D_{p}}+2\rho A\cdot \frac{\partial s}{\partial A}-2\rho A_{b}\cdot \frac{\partial s}{\partial A_{b}}=O,\\ \frac{\partial \psi }{\partial D_{b}}+2\rho A_{b}\cdot \frac{\partial s}{\partial A_{b}}=O. \end{array}$$
We omitted the fundamental equation about heat flux although it is obvious that ψ is a function of **L**, $D_{p}$, $D_{b}$ and heat flux **q**. Exploring the consequences from the Legendre transform, we can express the fundamental equations in terms of free energy as follows:(76)$$T=2\rho A\cdot \frac{\partial f}{\partial A},\;\theta \frac{\partial \psi }{\partial D_{p}}=T-T_{b},\;\theta \frac{\partial \psi }{\partial D_{b}}=T_{b},$$
where(77)$$T_{b}=2\rho A_{b}\cdot \frac{\partial f}{\partial A_{b}}.$$
Note that ∂ψ/∂L=O is assumed because we are interested in a material without viscous stress.

It can be easily recognized that the first of Equation (76) is the stress equation, the other two equations are flowing rules for $D_{p}$ and $D_{b}$. The flowing rules are dependent on the functional structure of dissipation function ψ. As before, let us assume that the dissipative function is a quadratic function as follows:(78)$$\psi =\psi_{\mathrm{heat}}\left(q,\;S_{f}\right)+\psi_{1}\left(D_{p},\;S_{f}\right)+\psi_{2}\left(D_{b},\;S_{f}\right),$$
where(79)$$\psi_{1}=\frac{\eta_{11}}{\theta }{\left(\mathrm{tr}D_{p}\right)}^{2}+\frac{\eta_{12}}{\theta }D_{p}:D_{p},\;\psi_{2}=\frac{\eta_{21}}{\theta }{\left(\mathrm{tr}D_{b}\right)}^{2}+\frac{\eta_{22}}{\theta }D_{b}:D_{b}.$$
Of course, the viscosities $\eta_{ik}$ are positive function of state variables, $S_{f}=\left(\theta ,\;A,\;A_{b}\right)$. Application of Equation (79) to Equation (76) gives(80a)$$D_{p}=\frac{1}{2\eta_{12}}\left[T-T_{b}-\zeta_{1}\mathrm{tr}\left(T-T_{b}\right)I\right],$$(80b)$$D_{b}=\frac{1}{2\eta_{22}}\left(T_{b}-\zeta_{2}\mathrm{tr}T_{b}\;I\right),$$
where(81)$$\zeta_{1}=\frac{\eta_{11}}{3\eta_{11}+\eta_{12}},\;\zeta_{2}=\frac{\eta_{21}}{3\eta_{21}+\eta_{22}}.$$
As before, if $\eta_{11}\gg \eta_{12}$ and $\eta_{21}\gg \eta_{22}$ then $\zeta_{1}=\zeta_{2}=\frac{1}{3}$ and both $D_{p}$ and $D_{b}$ are traceless tensors. This agrees with the most constitutive models based on the Kröner–Lee decomposition hypothesis but most assumptions used in previous models are derived from OVP and modeling of dissipation function.

The thermodynamic model from $S_{s}=\left(\varepsilon ,\;A,\;A_{b}\right)$ corresponds to the 4-element model, Figure 1d. The 4-element model is applicable to the plasticity with kinematic hardening. However, the above modeling cannot describe the material behavior such that unloading does not generate plastic deformation rate:(82)$$D_{p}=O\;\mathrm{for}\;\mathrm{unloading}.$$
According to Haupt [10], a mathematical representation of unloading may be given by(83)$$𝒫\equiv T^{*}:D<0,$$
where $T^{*}$ is a deriving stress for plastic deformation rate: $T^{*}=T-T_{b}$ for $D_{p}$ of Equation (80a) and $T^{*}=T_{b}$ for $D_{b}$ of Equation (80b). Then the fluidities of Equation (80) ($1/2\eta_{12}$ or $1/2\eta_{22}$) should be functions of 𝒫 which is proportional to **D**. Then, a contradiction encounters the dissipative function model of Equation (78). A solution to this problem will be discussed in the next section.

It is necessary to mention that the evolution equation of back stress, proposed by Armstrong and Frederick [15], is equivalent to(84)$$D_{b}\propto {\dot{e}}_{p}\left(T_{b}-\frac{\mathrm{tr}T_{b}}{3}I\right),\;{\dot{e}}_{p}=\sqrt{D_{p}:D_{p}}.$$
This means that $D_{b}$ can have non-zero value whenever $D_{p}\neq O$. This demands a special care on modeling dissipation function.

## 5. Rate-Independent Plasticity

Metal shows that stress is independent on deformation rate irrespective of yielding at moderate temperature range. A mathematical description of this dynamical behavior is given in Haupt [10] and Gurtin, Fried, and Anand [9]. According to the literature, a rate-independent plasticity obeys(85a)T=T(A),(85b)$$D_{p}\propto 𝒫\Theta \left(𝒫\right)T^{*},$$(85c)$$𝒫=D:T^{*},$$
where **A** is an elastic strain tensor, Θ(x) is the unit step function [Θ(x)=1 for x≥0 and Θ(x)=0] and $T^{*}$ is a deriving stress generating plastic flow. The deriving stress is a spatial tensor and in many cases, it is a deviatoric stress. Note that we define plastic dashpot as the one represented by Equation (85b). If $D_{p}$ is independent of **D** then the dashpot is called viscous dashpot.

For simplicity, we consider the 2-element model of Figure 1a, whose dashpot is rate-independent plastic one. The thermodynamic space is the one discussed in Section 3: $S_{s}=\left(\varepsilon ,\;A\right)$ (or $S_{f}=\left(\theta ,\;A\right)$). Then OVP with $R=-2A\cdot D_{p}$ gives(86a)$$T=2\rho A\cdot \frac{\partial f}{\partial A},$$(86b)$$\theta \frac{\partial \psi }{\partial D_{p}}=T,$$(86c)$$\frac{\partial \psi }{\partial D}=O$$
if we choose the dissipation function(87a)$$\psi =\psi_{\mathrm{heat}}\left(q,\;S_{f}\right)+\psi_{\mathrm{rlx}}\left(D_{p},\;D,\;S_{f}\right)+\psi_{e}\left(D,\;S_{f}\right),$$(87b)$$\theta \psi_{\mathrm{rlx}}\left(D_{p},\;D,\;S_{f}\right)=\eta_{1}\left(D,\;S_{f}\right){\left(\mathrm{tr}D_{p}\right)}^{2}+\eta_{2}\left(D,\;S_{f}\right)D_{p}:D_{p},$$(87c)$$\eta_{k}\left(D,\;S_{f}\right)=\frac{\mu_{k}\left(S_{f}\right)}{g\left(D:G\right)},\;k=1\;\mathrm{and}\;2.$$
Note that $\psi_{e}$ and *g* are functions of **D** that will be determined later using Equation (86c) and **G** is a tensor that is independent of both **D** and $D_{p}$. Substitution of Equation (87) to Equation (86b) results in(88)$$D_{p}=\frac{1}{2\eta_{2}}\left(T-\zeta \mathrm{tr}T\;I\right),\;\zeta =\frac{\eta_{1}}{3\eta_{1}+\eta_{2}}.$$
Equation (88) implies that the deriving stress is $T^{*}=T-\zeta \mathrm{tr}T\;I$ which could be a deviatoric stress if $\mu_{1}\gg \mu_{2}$.

Application of Equation (87) to Equation (86c) gives(89)$$\frac{\partial \psi_{e}}{\partial D}=\frac{\psi_{\mathrm{rlx}}\left(D_{p},\;D,\;S_{f}\right)g^{\prime }\left(D:G\right)}{g\left(D:G\right)}G,$$
where $g^{\prime }\left(x\right)=dg/dx$. For simplicity, assume that g(D:G) is a linear function such that g(x)=cxΘ(x), where *c* is a positive dimensionless constant. Then Equation (89) can be rewritten by(90)$$\frac{\partial \psi_{e}}{\partial D}=\{\begin{array}{cc} \frac{\mu_{1}\left(S_{f}\right){\left(\mathrm{tr}D_{p}\right)}^{2}+\mu_{2}\left(S_{f}\right)D_{p}:D_{p}}{{\left(cD:G\right)}^{2}}G & \mathrm{for}\;D:G\geq 0\\ O & \mathrm{otherwise} \end{array}.$$
With the help of Equation (87c), Equation (88) can be rewritten by(91)$$D_{p}=\frac{cD:G\Theta \left(D:G\right)}{2\mu_{2}}\left(T-\zeta \mathrm{tr}T\;I\right),\;\zeta =\frac{\mu_{1}}{3\mu_{1}+\mu_{2}}.$$
Since $D_{p}$ is a linear function of **D** and ζ is a function of only state variables, substitution of Equation (90) to Equation (89) lets us recognize that the right-hand side of Equation (89) is not dependent of **D**. Then we can construct $\psi_{e}$ as(92)$$\psi_{e}\left(D,\;S_{f}\right)=\{\begin{array}{ll} \frac{1}{4}\left[\frac{\mu_{1}}{\mu_{2}}{\left(\mathrm{tr}T^{*}\right)}^{2}+T^{*}:T^{*}\right]\frac{G:D}{\mu_{2}} & \mathrm{for}\;G:D\geq 0\\ 0 & \mathrm{otherwise} \end{array},$$
where(93)$$T^{*}=T-\zeta \mathrm{tr}T\;I.$$
Comparison of (85b) with Equation (90) determines the tensor **G** by the deriving stress $T^{*}$: $G=T^{*}$. Then Equation (90) can be rewritten by(94)$$D_{p}=\{\begin{array}{ll} \frac{cD:T^{*}}{2\mu_{2}\left(S_{f}\right)}T^{*} & \mathrm{for}\;D:T^{*}\geq 0\\ O & \mathrm{otherwise} \end{array}.$$
From Equation (87c), since *c* is dimensionless, $\eta_{k}$ has the dimension of viscosity and $D:T^{*}$ has the dimension of stress per time, $\mu_{k}$ must have the dimension of the square of stress. One of the simplest models for $\mu_{2}$ should be given by(95)$$\mu_{2}=\kappa T^{*}:T^{*},$$
where κ is a positive dimensionless constant.

One may want to introduce a yield condition to Equation (94) such that(96)$$D_{p}=\{\begin{array}{ll} \frac{cD:T^{*}}{2\mu_{2}\left(S_{f}\right)}T^{*} & \mathrm{for}\;D:T^{*}\geq 0\;\mathrm{and}\;𝒴(T^{*})\geq 0\\ O & \mathrm{otherwise} \end{array}.$$
where one of the simplest model for the yield function would be given by(97)$$𝒴\left(T^{*}\right)=T^{*}:T^{*}-\sigma_{o}^{2}.$$
Then Equation (96) can be rewritten by(98)$$D_{p}=\frac{𝒫\Theta \left(𝒫\right)\Theta \left(𝒴\right)}{2\mu_{2}\left(S_{f}\right)}T^{*}.$$
Here, the constant *c* is absorbed in $\mu_{2}$.

In order to derive $D_{b}$ thermodynamically satisfying Equation (84), the dissipation function must be remodeled in the same manner as before.

Let’s move to rate-independent model of plasticity which can be represented by the 4-element model in Figure 1d. The model is spanned by the thermodynamic space such as $S_{s}=\left(\varepsilon ,\;A,\;A_{b}\right)$ [or $S_{f}=\left(\theta ,\;A,\;A_{b}\right)$]. The evolution equations of the internal variables can be given by(99a)$$\frac{dA}{dt}=L\cdot A+A\cdot L^{T}-2A\cdot D_{p},$$(99b)$$\frac{dA_{b}}{dt}=D_{p}\cdot A_{b}+A_{b}\cdot D_{p}-2A_{b}\cdot D_{b}.$$
Application of Equation (99) to OVP yields the following stress equation:(100)$$T=2\rho A\cdot \frac{\partial f}{\partial A}.$$
Here, it is obvious that it is assumed that ∂ψ/∂L=O. For the rate-independence, we assume the mathematical form of dissipation function as follows:(101)$$\begin{array}{ll} \psi & =\psi_{\mathrm{heat}}\left(q,\;S_{f}\right)+\psi_{1}\left(D_{p},\;D,\;S_{f}\right)+\psi_{b}\left(D_{b},\;D_{p},\;S_{f}\right)\\ & \;+\psi_{e}\left(D,\;S_{f}\right)+\psi_{f}\left(D_{p},\;S_{f}\right), \end{array}$$
where(102)$$\begin{array}{l} \theta \psi_{1}=\eta_{11}{\left(\mathrm{tr}D_{p}\right)}^{2}+\eta_{12}D_{p}:D_{p},\\ \theta \psi_{b}=\eta_{21}{\left(\mathrm{tr}D_{b}\right)}^{2}+\eta_{22}D_{b}:D_{b},\\ \eta_{11}=\frac{\mu_{11}\left(S_{f}\right)}{g\left(D,\;S_{f}\right)},\;\eta_{12}=\frac{\mu_{12}\left(S_{f}\right)}{g\left(D,\;S_{f}\right)},\\ \eta_{21}=\frac{\mu_{21}\left(S_{f}\right)}{h\left(D_{p},\;S_{f}\right)},\;\eta_{12}=\frac{\mu_{22}\left(S_{f}\right)}{h\left(D_{p},\;S_{f}\right)}, \end{array}$$
Since OVP gives(103)$$\theta \left(\frac{\partial \psi_{1}}{\partial D_{p}}+\frac{\partial \psi_{b}}{\partial D_{p}}+\frac{\partial \psi_{f}}{\partial D_{p}}\right)=T-T_{b},\;\theta \frac{\partial \psi_{b}}{\partial D_{b}}=T_{b}=2\rho A_{b}\cdot \frac{\partial f}{\partial A_{b}},$$
Equation (102) results in(104)$$\begin{array}{l} 2\eta_{11}\mathrm{tr}D_{p}\;I+2\eta_{12}D_{p}=\frac{\psi_{b}}{h}\frac{\partial h}{\partial D_{p}}-\frac{\partial \psi_{f}}{\partial D_{p}}+T-T_{b},\\ 2\eta_{21}\mathrm{tr}D_{b}\;I+2\eta_{22}D_{b}=T_{b} \end{array}$$
The assumption for the stress equation, ∂ψ/∂L=O becomes(105)$$\frac{\partial \psi_{e}}{\partial D}=\frac{\psi_{1}}{g}\frac{\partial g}{\partial D}.$$

Without loss of generality, we can set the two auxiliary dissipation functions $\psi_{e}$ and $\psi_{f}$ by use of Equation (105) and(106)$$\frac{\partial \psi_{f}}{\partial D_{p}}=\frac{\psi_{b}}{h}\frac{\partial h}{\partial D_{p}}.$$
Then the flow rules are given by(107a)$$D_{p}=\frac{g\left(D,\;S_{f}\right)}{2\mu_{12}\left(S_{f}\right)}\left[T-T_{b}-\zeta_{1}\mathrm{tr}\left(T-T_{b}\right)\;I\right]$$(107b)$$D_{b}=\frac{h\left(D_{p},\;S_{f}\right)}{2\mu_{22}\left(S_{f}\right)}\left(T_{b}-\zeta_{2}\mathrm{tr}T_{b}\;I\right).$$
Now we can model $g\left(D,\;S_{f}\right)$ and $h\left(D_{p},\;S_{f}\right)$ in order that the constitutive equation obeys rate-independence:(108a)$$g\left(D,\;S_{f}\right)=𝒫\Theta \left(𝒫\right),$$(108b)$$h\left(D_{p},\;S_{f}\right)={\dot{e}}_{p},$$(108c)$${\dot{e}}_{p}=\sqrt{D_{p}:D_{p}}.$$
Of course, as before, we define 𝒫 as(109)$$𝒫=D:T^{*},\;T^{*}=T-T_{b}-\zeta_{1}\mathrm{tr}\left(T-T_{b}\right)\;I.$$
Note that the driving stresses in Equation (107) are tensor-valued functions of $S_{f}=\left(\theta ,\;A,\;A_{b}\right)$. Then Equation (99) is rate-independent evolution equations and so is stress. If one may want to introduce yield criterion to the flow rule then $g\left(D,\;S_{f}\right)$ is modified to(110)$$g\left(D,\;S_{f}\right)=𝒫\Theta \left(𝒫\right)\Theta \left(𝒴\right).$$

This model can be considered as a model of rate-independent plasticity with kinematic hardening. Equation (99b) with Equations (107b) and (108b) is a large-deformation version of the evolution equation of back stress proposed by Armstrong and Frederick [15].

## 6. Viscoplasticity

The stress in viscoelasticity depends on both the deformation path and the deformation rate, which is also the case for viscoplasticity. So what is the difference between viscoelasticity and viscoplasticity? Some may think about the presence or absence of a yield surface, but if we consider the endochronic model [34], plasticity can be expressed even without a yield surface. The author agrees with the classification proposed by Haupt [10]. Haupt added a condition that viscoplasticity shows a hysteresis loop of fully relaxed stress, whereas viscoelasticity does not. Haupt’s classification can be visualized by Figure 2.

Since rate-independent plasticity does not show stress relaxation, the hysteresis loop of stress is identical to that of fully relaxed stress as shown in Figure 2b. The stress-strain curve of fully relaxed stress corresponds to the curve generated by $T_{2}$ and $A_{2}$ of the 3-element model I represented in Figure 1b. Therefore, one of the simplest viscoplastic model would be the parallel combination of the 2-element model with viscous dashpot and the 2-element model with rate-independent plastic dashpot (Figure 3a). Since the fully relaxed stress of the Maxwell element with viscous dashpot is zero, the total fully relaxed stress of the model is that of the Maxwell element with rate-independent plastic dashpot. In general, viscoplastic model can be constructed by a parallel connection of viscoelastic and rate-independent plastic models. Therefore, the Cauchy stress of viscoplastic model is the sum of the stresses of viscoelastic and rate-independent plastic models:(111)$$T=T_{\mathrm{VE}}+T_{\mathrm{RI}},$$
where $T_{\mathrm{VE}}$ is the stress of viscoelasticity, which has rate-dependence and $T_{\mathrm{RI}}$ is the stress of rate-independent plasticity.

If one may want to introduce kinematic hardening then the 6-element model of viscoplasticity shown in Figure 3b could be chosen. Note that the dashpots represented by $D_{\mathrm{RI}}$ and $D_{b}$ are driven by $T_{\mathrm{RI}}-T_{b}$ and $T_{b}$, respectively, and they are rate-independent dashpots. On the other hand, the dashpot represented by $D_{\mathrm{VE}}$ is a viscous dashpot and driven by $T_{\mathrm{VE}}$.

Figure 3 shows that the 4-element model of viscoplasticity has two internal variables $A_{\mathrm{VE}}$ and $A_{\mathrm{RI}}$. Both internal variables are assumed as symmetric and positive definite tensors. Since they are related with $T_{\mathrm{VE}}$ and $T_{\mathrm{RI}}$, their evolution equation can be given by(112)$$\begin{array}{l} \frac{dA_{\mathrm{VE}}}{dt}=L\cdot A_{\mathrm{VE}}+A_{\mathrm{VE}}\cdot L^{T}-2A_{\mathrm{VE}}\cdot D_{\mathrm{VE}},\\ \frac{dA_{\mathrm{RI}}}{dt}=L\cdot A_{\mathrm{RI}}+A_{\mathrm{RI}}\cdot L^{T}-2A_{\mathrm{RI}}\cdot D_{\mathrm{RI}}, \end{array}$$
Then OVP gives the stress equation such that(113)$$T=T_{\mathrm{VE}}+T_{\mathrm{RI}},\;T_{\mathrm{VE}}=2\rho A_{\mathrm{VE}}\cdot \frac{\partial f}{\partial A_{\mathrm{VE}}},\;T_{\mathrm{RI}}=2\rho A_{\mathrm{RI}}\cdot \frac{\partial f}{\partial A_{\mathrm{RI}}}.$$
Using the methodology learned in the previous sections, $D_{\mathrm{VE}}$ and $D_{\mathrm{RI}}$ should obey the following flow rule:(114)$$D_{\mathrm{VE}}=\frac{1}{2\eta \left(S_{f}\right)}\left(T_{\mathrm{VE}}-\zeta \mathrm{tr}T_{\mathrm{VE}}\;I\right),\;D_{\mathrm{RI}}=\frac{𝒫\Theta \left(𝒫\right)\Theta \left(𝒴\right)}{\mu \left(S_{f}\right)}\left(T_{\mathrm{RI}}-\zeta \mathrm{tr}T_{\mathrm{RI}}\;I\right),$$
where $𝒫=D:\left(T_{\mathrm{RI}}-\zeta \mathrm{tr}T_{\mathrm{RI}}\;I\right)$ and $𝒴(T_{\mathrm{RI}})$ is a suitable yield function. Note that one may model the specific free energy by $f=f_{\mathrm{VE}}\left(\theta ,\;A_{\mathrm{VE}}\right)+f_{\mathrm{RI}}\left(\theta ,\;A_{\mathrm{RI}}\right)$.

If compressibility should be considered then we need the evolution equation of density which is given by(115)$$\frac{d\rho }{dt}=-\rho \mathrm{tr}D.$$
Note that our constitutive theory is designed to calculate stress when velocity gradient is given. Then Equation (115) can determine ρ at an arbitrary time if **D** and the initial condition are given.

It is straightforward that the 6-element model of viscoplasticity in Figure 3b can be constructed by combining the results in previous sections.

## 7. Conclusions

It is showed that the application method of OVP, which was proposed by Cho and Lee [5], can replace a large set of assumptions in conventional theories of viscoelasticity and viscoplasticity by a smaller set of assumptions of irreversible thermodynamics. Therefore, the constitutive equations from the OVP are not new constitutive equations and this paper shows that the OVP can derive the existing constitutive equations without ad hoc assumptions that have been adopted in traditional modeling. It could be said that the previous constitutive equations show thermodynamical consistency even though they are not based on thermodynamics or partially adopt thermodynamic principles such as the non-negativity of entropy production rate.

The Kröner–Lee decomposition is based on the hypothesis that there is an intermediate configuration. In order to ensure the uniqueness of this decomposition theory, rotational motion should be involved only in the elastic deformation gradient. Since the Kröner–Lee decomposition is useful in providing the evolution equation of elastic strain, several subsidiary hypotheses are needed to support it. However, the OVP in this paper can construct the whole constitutive equation without such hypothesis.

Most conventional constitutive theories assume that plastic deformation is incompressible and that the yield is independent of hydrostatic pressure, which agree with the experiments on metals. However, the experiments on polymeric solid do not obey the hypothesis which plays important role in the conventional theories for flowing rule. However, the OVP in this paper does not have to concern the hypothesis because the modeling of dissipation function can describe both metal and polymeric solid.

Although the principle of virtual power suggested by Gurtin and coworkers provides stress equation which is the relation between stress and elastic strain and other deformation measures, most conventional theories depends on some hypothesis on stress equation. The OVP naturally provides the stress equation thermodynamically without considering the principle of material indifference which is necessary for the principle of virtual power.

The flowing rule of conventional constitutive theories is the connection between plastic deformation rate tensor and stress measures, which are also based on intuitive hypothesis. However, the OVP naturally provides the flowing rule, too.

## Figures and Tables

**Figure 1 entropy-27-00055-f001:**
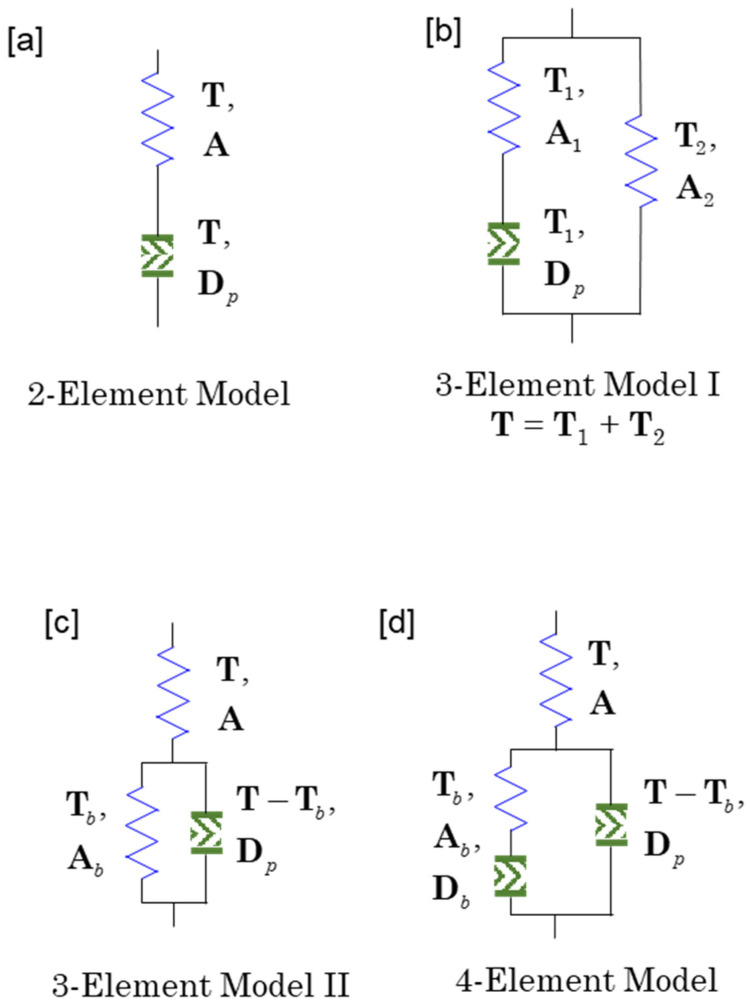
The fundamental components for modeling viscoelasticity and viscoplasticity. (**a**) A 2-element model consisting of a series connection of an elastic spring and an inelastic dashpot, (**b**) a 3-element model I with an additional elastic element in parallel, (**c**) a 3-element model II for explaining back stress, (**d**) a 4-element model incorporating both inelastic behavior and back stress.

**Figure 2 entropy-27-00055-f002:**
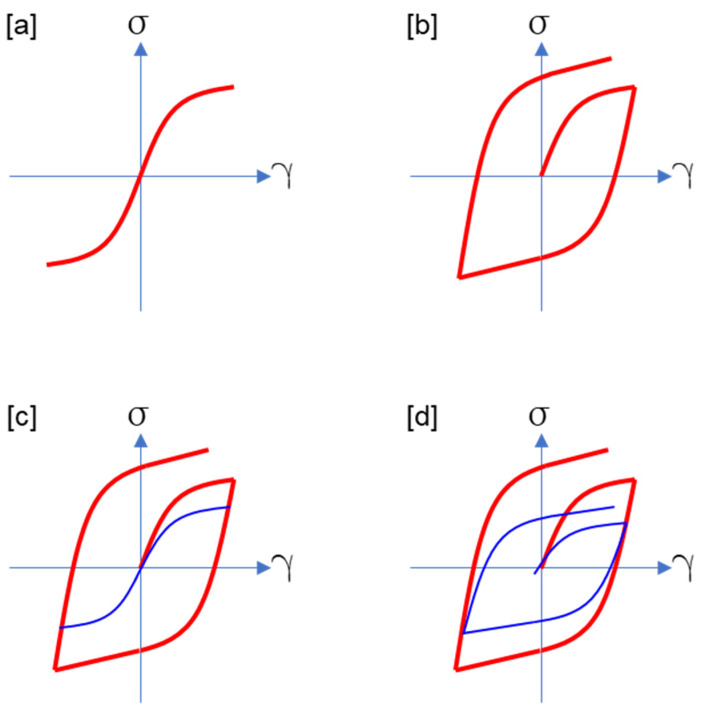
Classification of materials by cyclic loading and relaxation. The thick curves (red) are the stress-strain curves for cyclic strain and the thin curves (blue) represent the fully relaxed stress which can be obtained from the relaxation at each points of the thick curves. (**a**) elasticity; (**b**) rate-independent plasticity; (**c**) viscoelasticity; (**d**) viscoplasticity.

**Figure 3 entropy-27-00055-f003:**
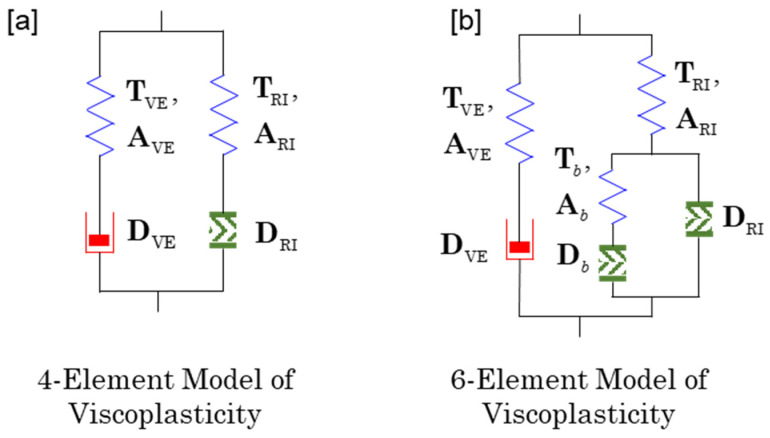
Extended configurations for viscoplastic modeling. (**a**) A 4-element model combining viscoelastic and rate-independent behaviors in parallel, (**b**) a 6-element model integrally describing viscoelastic and viscoplastic behaviors.

## Data Availability

The data presented in this study are available on request from the corresponding authors.

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
