# Peer review of "Unified Analysis of Viscoelasticity and Viscoplasticity Using the Onsager Variational Principle"

_entropy, 2025, doi:10.3390/e27010055_

Round 1

Reviewer 1 Report

Comments and Suggestions for Authors

1st Report on “Unified Analysis of Viscoelasticity and Viscoplasticity using Onsager Variational Principle”

The author contracts the OVP for finite strain theory. That provides the systematic derivation of governing equations such as the stress equations. The previous approach of finite strain theory includes several assumptions and multiplicative decomposition; the author shows that these are obtained from OVP systematically in this work. The author further shows how to apply the general framework to multi-element models for viscoelasticity and viscoplasticity.

The paper has some careless issues. I noticed that many equations were misprinted, in particular, most of “(“ and “)” disappeared in the pdf I received. I do not know which the author or editor’s process had a problem. Fixing them is needed to judge the correctness of equations.

To my knowledge, though OVP was used for small strain theory by Doi and others, OVP for finite strain theory is still being constructed. Hence, the research might be important to understand the large deformation of various realistic elastic materials. However, I am not sure about the research position of this work because the author did not describe what the previous approaches are using OVP to viscoelastic material like Doi’s work, and I cannot confirm what problem in the previous approach was and how this work solved the problems. Although the author mentioned Doi’s framework shortly, no Doi’s papers are on the reference list.

I believe the current manuscript will satisfy the criteria of the Entropy after some revision addressing the following comments.

(i)As mentioned above, it is needed to describe the author's progress sharply from the previous works, e.g., Doi’s work on viscoelastic media. I noticed the author discussed the difference of Doi’s framework and the author’s framework in general in Ref[20]. However, this discussion is also needed to highlighted even in this paper focusing on viscoelastic media, and I recommend citing Doi’s papers[M. Doi, Onsager principle in polymer dynamics, Progress in Polymer Science 112, 101339 (2021), Doi M (2013) Soft matter physics. Oxford University Press, Oxford].

(ii)Below Eq(16), the author mentions that Doi’s OVP cannot provide the stress equation because Doi omits internal energy ϵ. However, the author's stress equation Eq(17)_1 does not include internal energy. That is inconsistent with the author's explanation of Doi’s OVP.

And Ref[20] and Doi’s paper[M. Doi, Onsager principle in polymer dynamics, Progress in Polymer Science 112, 101339 (2021)] mention that Doi’s modification suits the system in isothermal with a thermal bath. That indicates Doi’s OVP works well even for material if the system isothermal. Could the author explain Doi’s OVP deeply?

(iii) In section III, the author presents their main results. However, the author’s contribution was not sharp enough for the reader's understanding. Could the author reconstruct section III to clarify which part is the review of Kröner-Lee decomposition and which part is the author’s contribution using OVP?

(iv)OVP has been constructed in an irreversible process under the linear response. So, the dissipation part should be a linear response, and the dissipation function is quadratic of fluxes. Hence, the viscous or plastic part of the material should be in the linear response. For the elastic part, on the other hand, the author considers nonlinear response and geometrical nonlinearity coming from finite strain theory. Could the author summarize linear and nonlinear relations in their OVP.

(v)Eq.(36), the author introduces the nonlinear material constant ϵ, which is used for internal energy, too. Could the author use a different notation?

(vi)I point out the following equation loss “(“ and “).” I cannot extract all of the errors. Could the author recheck all of the equations?

Eqs.(9)(39)(40)(41)(43)(46)(47)(48)(78)(85)(87)(89) etc….

In line equations: below Eq(9), below Eq(22), above Eq(24), below Eq(32), below Eq(35), below Eq(85),etc…

Author Response

I have uploaded the responses to the reviewers as a PDF file.

Reviewer 2 Report

Comments and Suggestions for Authors

See the attached file for comments. 

Comments on the Quality of English Language

Fine but it should be further polished. 

Author Response

(The authors gave the same response as above.)

Round 2

Reviewer 1 Report

Comments and Suggestions for Authors

See the attached PDF file.

Author Response

The answer is attached as a PDF file.

Reviewer 2 Report

Comments and Suggestions for Authors

The authors have revised the manuscript significantly and all my concerns have been fully addressed. I think it is now in a much better form, ready to be published. 

Author Response

The answer is attached as a PDF file.
